# Silk Gland Factor 1 Plays a Pivotal Role in Larval Settlement of the Fouling Mussel *Mytilopsis sallei*

**DOI:** 10.3390/biology13060417

**Published:** 2024-06-05

**Authors:** Jian He, Zhixuan Wang, Zhiwen Wu, Liying Chen, Jianfang Huang

**Affiliations:** 1College of Geography and Oceanography, Minjiang University, Fuzhou 350108, China; hejian@mju.edu.cn; 2Fuzhou Institute of Oceanography, Minjiang University, Fuzhou 350108, China; 3College of Ocean & Earth Sciences, Xiamen University, Xiamen 361102, China; zxwang@stu.xmu.edu.cn (Z.W.); wuzhiwen0717@foxmail.com (Z.W.); 22320190154004@stu.xmu.edu.cn (L.C.)

**Keywords:** biofouling, *Mytilopsis sallei*, larval settlement, SGF1, virtual screening

## Abstract

**Simple Summary:**

Marine biofouling is the undesirable accumulation of marine organisms onto man-made surfaces, which causes severe economical and environmental consequences. In recent years, using natural products as environmentally friendly antifouling agents to combat foulers has been a promising approach. However, the search for effective antifouling agents is hampered by the lack of well-defined conserved molecular targets responsible for regulating the larval settlement in fouling organisms. In this study, an in silico approach was used to screen the natural compounds’ libraries against silk gland factor 1 (SGF1), which is responsible for the larval settlement of the fouling mussel *Mytilopsis sallei*. It was found that the targeted binding compounds against SGF1 could significantly affect the larval settlement, foot proteins’ gene expression, and byssus secretion of adults in *M. sallei*. The laboratory bioassay results are promising, even though field trials are necessary to confirm the in silico results. Hence, this study paves the way for new experimental studies to be carried out for finding environmentally friendly antifouling agents by using SGF1-targeted compounds.

**Abstract:**

Most fouling organisms have planktonic larval and benthic adult stages. Larval settlement, the planktonic–benthic transition, is the critical point when biofouling begins. However, our understanding of the molecular mechanisms of larval settlement is limited. In our previous studies, we identified that the AMP-activated protein kinase-silk gland factor 1 (AMPK-SGF1) pathway was involved in triggering the larval settlement in the fouling mussel *M. sallei*. In this study, to further confirm the pivotal role of SGF1, multiple targeted binding compounds of SGF1 were obtained using high-throughput virtual screening. It was found that the targeted binding compounds, such as NAD^+^ and atorvastatin, could significantly induce and inhibit the larval settlement, respectively. Furthermore, the qRT-PCR showed that the expression of the foot proteins’ genes was significantly increased after the exposure to 10 μM NAD^+^, while the gene expression was significantly suppressed after the exposure to 10 μM atorvastatin. Additionally, the production of the byssus threads of the adults was significantly increased after the exposure to 10–20 μM of NAD^+^, while the production of the byssus threads was significantly decreased after the exposure to 10–50 μM of atorvastatin. This work will deepen our understanding of SGF1 in triggering the larval settlement in mussels and will provide insights into the potential targets for developing novel antifouling agents.

## 1. Introduction

Marine biofouling is the undesirable accumulation of marine organisms on an artificially immersed surface in seawater, which causes severe economical as well as environmental consequences [1,2]. The accumulation on the vessel’s hull results in damage to the rudder and propulsion systems, which leads to an increasing drag of up to 60%, as well as a fuel consumption increase by 40% per year due to biofouling [3,4]. In addition, fouling organisms that gregariously settle on submerged surfaces severely clog the circulation piping in the energy infrastructure, which decreases the heat exchange efficiency and affects the operation of generators [5]. Moreover, the metabolites produced by the fouling organisms can induce the corrosion of the submerged facilities [6,7]. Therefore, biofouling prevention has become the most challenging issue in marine industries.

The traditional antifouling technologies have relied on the incorporation of toxic biocidal compounds, such as heavy metals, to kill fouling organisms [8,9,10]. However, legislative changes in recent years have signaled the gradual outlawing of many such compounds, which calls for environmentally friendly antifouling methods, such as the use of natural biological products with antifouling activity as antifouling agents [11,12]. Most of the previous studies on antifouling agents have mainly focused on the antifouling activity, while the underlying molecular mechanisms of antifouling are yet little known, which is mainly because the adhesion mechanism of fouling organisms is not well-understood. It is difficult to develop high-efficiency and environmentally friendly antifouling agents to combat foulers with the lack of information on marine organisms’ adhesion. Therefore, the identification of the key attachment mechanisms is both industrially and ecologically significant [13].

Most fouling organisms have planktonic larval and benthic adult stages [14,15]. When the planktonic individuals are fully developed, they begin to swim toward the bottom to search for appropriate sites for settlement. Then, the individuals will secrete the adhesives for attachment and metamorphose into benthic juveniles. Since the juveniles or adults lose motility or show reduced motility, the planktonic–benthic transition is the critical point when the fouling of surfaces begins [16]. The attachment and metamorphosis (both processes are referred to, in the present paper, as ‘settlement’) are complex processes involving many important morphological, physiological, and biochemical changes [17,18]. So far, a number of signaling systems have been shown to be involved in triggering the larval settlement in marine invertebrates, such as neurotransmitter signaling [17,19], mitogen activated protein kinase signaling [20,21], and the protein kinase C signaling pathway [21]. However, our understanding of the underlying molecular mechanisms of fouling organism settlement is still limited, especially for how the signaling pathways regulate the adhesives’ secretion.

Dreissenid mussels are well-known invasive species and economic pests in aquatic ecosystems. They can settle gregariously on submerged manmade structures, causing serious biofouling problems [22]. Recently, we identified that the mechanosensitive transient receptor potential channel was involved in the substrate exploration for larval settlement in the dreissenid mussel *Mytilopsis sallei* [23]. Further, it was shown that the transient receptor potential channel triggered the larval settlement through the calmodulin-dependent protein kinase kinase β/AMP-activated protein kinase/silk gland factor 1 (CaMKKβ-AMPK-SGF1) pathway [23]. It is worth noting that SGF1 plays an important role in activating the fibroin gene expression in the silkworm [24,25]. The mussels attach to the substrates via the byssus, which is a silk-fibroin-like structure [26,27,28]. Therefore, we hypothesize that the transcription factor SGF1 plays a critical role in activating the gene expression of foot proteins (the main component of byssus). However, no specific activator or inhibitor of SGF1 has been reported currently, which obstructs the deeper functional research of SGF1.

In this study, multiple targeted binding compounds of SGF1 were obtained using high-throughput virtual screening. Further, the mussel responses to the compounds, including the larval settlement, foot proteins’ gene expression change, and byssus secretion of adults, were investigated. This work will deepen our understanding of the molecular mechanisms of fouling organism settlement and will provide insights into the potential targets for developing novel antifouling agents.

## 2. Materials and Methods

### 2.1. Homology Model

The 3D structure of SGF1 was developed using homology modelling. The query sequence was obtained from the transcriptome data of *M. sallei* [29]. A template for this sequence was identified using Basic Local Alignment Search Tool (BLAST+2.15.0). The crystal structure of Human FOXC2 (PDB ID: 6LBM) was selected as the template and had an overall sequence similarity of 73.47%. A sequence alignment was performed using the ClustalW (version 2.0) server tool [30]. Homology models were generated using SWISS-MODEL (https://swissmodel.expasy.org, accessed on 1 April 2024).

### 2.2. Virtual Screening

Virtual screening was performed by using Schrödinger Maestro 11.4 software against three libraries of compounds, including Discovery Diversity Set 50 (MedChemExpress, Shanghai, China, containing 50200 compounds), Life Chemicals 50K Diversity Library (MedChemExpress, Shanghai, China, containing 50200 compounds), and MCE Bioactive Compound Library (MedChemExpress, Shanghai, China, containing 15700 compounds). Schrodinger Glide was used for molecular docking. First, the high-throughput screening (HTVS) mode in Glide was subjected to screen the top-ranked 10% of compounds from the three libraries. Then, the standard (SP) mode was subjected to screen the top-ranked 10% of compounds in the second round of screening, and the extra precision (XP) mode was used to obtain the ranking of the compounds in the third round of screening. Finally, the binding forces of SGF1 and compounds and compound structures were manually rechecked, and the top-ranked 100 compounds were selected.

### 2.3. Spawning Induction and Larval Culture of M. sallei

Gravid adults of *M. sallei* (shell length, 20–30 mm) were collected from the Yundang Lagoon, Xiamen, China (24°29′ N, 118°04′ E). Spawning induction and larval culture were carried out in the laboratory following our previously published protocol [31] with some modifications. Briefly, about 500 individuals were exposed in air overnight and then placed into a 30 L plastic tank containing warm filtered seawater (FSW; UV-treated; temperature 32 °C). During spawning induction, the seawater was aerated vigorously. Fertilized eggs were collected with the mesh nets and incubated at a density of 20–30 individuals mL^−1^ in FSW at 27 ± 1 °C in the dark. After 24 h of incubation, the D-shaped veliger larvae were collected and transferred to 20 L incubation tanks containing fresh FSW. Larvae were incubated at a density of 3–5 individuals mL^−1^ and were fed with the single-celled golden algae *Dicrateria zhanjiangensis* (Chrysophyta) at a concentration of 1.0–5 × 10^4^ cells mL^−1^. The FSW was aerated gently and changed daily. After 6–8 days of larval incubation, larvae were able to swim in the seawater and crawl on the substrate with the feet for short intervals, which indicates that the pediveliger stage was reached. Pediveligers that were capable of settlement were used for the bioassays.

### 2.4. Bioassays of Larval Settlement in Response to Compounds

Bioassays were conducted in sterile six-well polystyrene Petri plates following the previous study [32]. Briefly, 0.4, 1, 2, 4, and 10 mol/L of test solution were prepared by dissolving the compounds in DMSO, respectively. Then, 50 μL of test solution, 9.95 mL of FSW, and 30–40 pediveliger larvae were added into each well of the Petri plate. Three replicates were set up for each treatment. FSW with 0.5% DMSO was used as a control. The six-well Petri plates were maintained at 27 °C in the dark. After 48 h of incubation, larval settlement and mortality were observed through a Leica inverted microscope (DM IL LED, Wetzlar, Germany). Larval attachment was indicated by crawling on the substrate with a foot or attaching with byssus; larval metamorphosis was confirmed by loss of the velum, or the appearance of mature gill filaments. Larval mortality was confirmed by the tissue inside the shell becoming ulcerated and no signs of movement of the velum, foot, or gut. In this study, regarding attachment and metamorphosis, both processes are referred to as settlement.

### 2.5. Foot Proteins’ Gene Expression Analysis by qRT-PCR

To demonstrate that the expression of foot proteins can been influenced by SGF1, the gene expression changes regarding foot proteins in response to the targeted binding compounds of SGF1 were investigated. Approximately 2000 pediveliger larvae were collected and transferred into a 250 mL beaker containing 200 mL filtered seawater. Then, the compound dissolving in DMSO was added to achieve a concentration of 10 µM. After 12 h of culture at 27 °C in the dark, the larvae of each beaker were collected and washed with 0.01 M PBS. The sample was frozen in liquid nitrogen rapidly and stored at −80 °C until processing. Three replicates were set up for each treatment. The larvae without exposure to the compounds were collected as control.

The qRT-PCR analysis was carried out following our published protocol [26] with some modifications. Briefly, total RNA of the samples with three replicates was extracted with Trizol reagent (Invitrogen, Carlsbad, CA, USA) following the manufacturer’s instructions. Total RNA (1 µg) from each sample was used as the template for the reverse transcription reaction with a PrimeScript™ RT reagent Kit (Takara, Shiga, Japan). The first-strand cDNA was synthesized and used as the template for further PCR analysis. According to the ORF sequences of foot proteins’ genes (Appendix A) from the transcriptome data [29], matching oligonucleotide primers (Appendix A) were designed. PCR was performed with a program of 95 °C for 5 min, 45 cycles of 95 °C for 20 s, and annealing at 60 °C for 20 s, followed by 72 °C for 30 s. The 2^−ΔΔCt^ Method [33] was utilized to evaluate the relative expression level of foot proteins’ genes with *β-actin* as the internal control.

### 2.6. Bioassays of Byssus Thread Secretion in Response to Compounds

The byssus thread secretion in response to compounds was tested by following the methods of Wang et al. [34] with some modifications. Briefly, *M. sallei* (shell length 8–12 mm) was collected from the Yundang Lagoon and then kept in an aquarium with aerated seawater at 27 ± 1 °C for 1–2 days. The byssal threads of each mussel were gently cut off with sharp scissors, and 2, 4, and 10 mol/L of NAD^+^ and atorvastatin were prepared by dissolving the compounds in DMSO, respectively. Then, one mussel, 20 μL of test solution, and 3.98 mL of FSW were added into each well of a 12-well polystyrene Petri plate. Ten replicates were set for each treatment and the control (0.5% DMSO in FSW). After 24 h of incubation, 4 mL of Coomasie blue was added into each well to stain the adhesive plaque for 1 h. After washing thoroughly, the adhesive plaque stained by Coomasie blue was counted and photographed.

### 2.7. Statistical Analysis

Results were analyzed with SPSS 26.0 software. Prior to analysis, all the data expressed as percentages of larval settlement and mortality were arcsine-transformed. For the bioassays of larval settlement in response to compounds, and byssus thread secretion in response to compounds, one-way analysis of variance (ANOVA) was performed with a Dunnett’s post hoc test for multiple comparisons of treatment means with the control. For the gene expression analysis, ANOVA was performed with Tukey’s test for multiple comparisons of treatment means. All the significance levels were set at *p* < 0.05.

## 3. Results

### 3.1. Homology Model and Virtual Screening

After the literature search, the full-length crystal structure of SGF1 had not yet been analyzed, and there was no three-dimensional structure of SGF1 predicted by AlphaFold. The SWISS-MODEL software (https://swissmodel.expasy.org, accessed on 1 April 2024) was used to predict the protein structure of SGF1. It showed that the optimal modeling template was Human FOXC2 (PDB ID 6LBM). The modeled sequence length was 157–253 amino acids, and the sequence similarity was as high as 73.47% (Figure 1A).

To determine the active pocket of SGF1, the modeled SGF1 and the template protein (Human FOXC2) were superimposed. As shown in Figure 1B. the two proteins were highly overlapping. It was calculated that the superimposed RMSD value was 0.072 Å. Therefore, the identification of the active pocket of SGF1 could refer to the binding sites of the Human FOXC2 to DNA. According to the literature report, there were three binding patterns of Human FOXC2 to DNA, one of which showed the optimal active sites for accommodating small-molecule compounds (Figure 1C) [35]. By sequence alignment, the amino acids in the active sites of SGF1 were LEU180/SER210/ASN203/HIS207. Then, the targeted binding compounds of SGF1 were screened based on these active sites.

The high-throughput screening (HTVS) mode in the Glide module was used to screen the small-molecule compounds from the compound libraries. In the docking study, a lower docking score indicates higher binding efficiency [36]. A total of 116100 compounds were analyzed. Based on the docking scores, the top-ranked 10 compounds were selected, with the docking values ranging from −9.0 to −12.0 (kcal/mol) (Table 1), and the information for the top-ranked 100 compounds is shown in Appendix A.

To demonstrate the effectiveness of virtual screening, the compound levomefolate (HY-17383) was further selected for analyzing the binding pattern with SGF1 (Figure 2). A total of four hydrogen bonds were formed: the carboxyl group acted as a hydrogen bond receptor to form a hydrogen bond with LYS217 at a distance of 2.0 Å; NH2 acted as a hydrogen bond donor to form a hydrogen bond with SER229 at a distance of 2.5 Å; and the amide bond in the ring formed two hydrogen bonds with SER229 at a distance of 2.0 Å. In addition, the carboxyl group could also form two ionic bonds with magnesium ions and one metal chelate cooperation. These results suggest that there was a strong binding force between levomefolate and SGF1, indicating that the high-throughput virtual screening was effective for obtaining the targeted binding compounds of SGF1.

### 3.2. Larval Settlement in Response to the Targeted Binding Compounds of SGF1

To confirm that the active form of SGF1 can affect the larval settlement of *M. sallei*, the larval responses to the top-ranked ten targeted binding compounds of SGF1 were examined (Figure 3). No effects on the settlement of *M. sallei* larvae were observed for levomefolate, folinic acid, levoleucovorin, PSMA-11, 1,3,6-Tri-O-galloyl-beta-D-glucose, and 1,2,3,6-Tetragalloylglucose at concentrations of 2–50 μM after a 48 h exposure compared to the control. Significant inducing effects on the larval settlement were observed for NAD^+^, cyclic-di-GMP, and S-adenosyl-DL-methionine at concentrations of 10–20 μM after a 48 h exposure. For NAD^+^, significant inducing effects on the larval settlement of *M. sallei* larvae were found at concentrations from 5 to 20 μM. In the case of cyclic-di-GMP, the treatments at 10–50 μM concentrations induced larval settlement significantly. For S-adenosyl-DL-methionine, significant inducing effects on the larval settlement were found at concentrations from 10 to 20 μM. It should be noted that, when the concentration of the NAD^+^ and S-adenosyl-DL-methionine reached above 50 μM, acute toxicity was observed on the *M. sallei* larvae. Meanwhile, significant inhibitive effects on the larval settlement were observed compared to the control for atorvastatin at concentrations of 5–50 μM after a 48 h exposure. Taken together, these results strongly suggest that SGF1 is important in mediating the larval settlement of *M. sallei*.

The mussels’ adhesion is initiated by the foot proteins forming the final adhesive plaque on the substrates [37,38]. To demonstrate that the expression of foot proteins can be influenced by SGF1, the gene expression changes regarding the foot proteins in response to the targeted binding compounds of SGF1 at the concentration of 10 μM were investigated. As shown in Figure 4, NAD^+^ significantly increased the gene expression of foot protein 2 (FP2), FP3, FP4, and FP5 compared to the control; cyclic-di-GMP significantly increased the gene expression of FP2 and FP5; S-adenosyl-DL-methionine significantly increased the gene expression of FP4 and FP5. Meanwhile, the gene expression of FP2, FP3, FP4, and FP5 was significantly suppressed compared to the control after exposure to 10 μM atorvastatin. No effect on the gene expression in response to all the compounds was observed for FP1 and FP6. These results suggest that the activity of SGF1 could influence the expression of foot proteins, including FP2, FP3, FP4, and FP5.

### 3.3. Byssus Thread Secretion in Response to the Targeted Binding Compounds of SGF1

To demonstrate that SGF1 plays a vital role in the secretion of foot proteins, the numbers of byssus threads produced by each mussel in response to NAD^+^ and atorvastatin were counted. As shown in Figure 5, the numbers of byssus threads produced by the *M. sallei* exposure to 10 and 20 μM NAD^+^ were 36.1 and 43.6, respectively, which were significantly increased compared with the control (21.0). In contrast, the production of byssus threads was significantly decreased after the exposure to 10–50 μM of atorvastatin. The results indicate similar trends with the larval settlement and foot proteins’ gene expression in response to NAD^+^ and atorvastatin.

## 4. Discussion

The activation of chemoreceptors or mechanoreceptors in the sensory cells responsible for settlement leads to the inducement of the signal transduction pathway, which results in the secretion of the adhesive protein and subsequently to the larval settlement [14,39]. By influencing the biochemical pathway, it would be possible to affect the larval settlement. Despite vigorous efforts to find new antifouling compounds, information regarding the mechanisms of antifouling is still scarce, which is hampered by the lack of well-defined conserved molecular targets. SGF1 is a FOXA transcription factor that contains an evolutionarily conserved forkhead or winged-helix DNA-binding domain [25,40], which plays a critical role in activating fibroin gene expression [24]. Hence, we hypothesize that SGF1 can be used as a molecular target for the larval settlement in mussels.

The mussels’ adhesion is initiated by the foot proteins (FPs) forming the final adhesive plaque on the substrates [37,38]. Six major types of foot proteins have been identified from mussels, including FP1, FP2, FP3, FP4, FP5, and FP6. FP1 covers the outer layer of the adhesive plaque and works as a protective cuticle [41]. FP2 is present in the inner part of the plaque and works as a major structural component [41]. FP3 and FP5 are located at the interface of the adhesive plaque and substrate surface and play a vital role in substrate adhesion [27,37]. The location of FP4 is between the precursor collagen in the distal end and FP2, where it acts as a coupling agent in the byssal thread and adhesive plaque [41]. FP6 is close to the surface of the adhesive plaque and bulk plaque proteins, which may provide a link between the DOPA-rich surfaces and bulk foot proteins [41]. Although extensive progress has been made, the adhesion mechanism is not fully understood, especially how the foot proteins are synthesized and secreted. This study showed that the expression of foot proteins, including FP2, FP3, FP4, and FP5, can be influenced by the targeted binding compounds against SGF1. It is reasonable to hypothesize that SGF1 plays an important role in regulating the synthesis of foot proteins for adhesion.

Nicotinamide adenine dinucleotide (NAD^+^) is an important electron carrier for the electron transport chain, ATP production, and other aspects of cellular metabolism [42]. So far, there are few reports on the effect of NAD^+^ on the larval settlement or metamorphosis in marine invertebrates. Nicotinamide, which serves as a building block for the synthesis of NAD^+^, was reported to stimulate the larvae of the polychaete *Capitella teleta* to settle and metamorphose [43]. It was suggested that the larvae of *C. teleta* responded to nicotinamide via a chemosensory pyridine receptor. Interestingly, the *C. teleta* larvae did not stimulate settlement or metamorphosis when exposed to NAD^+^, which might indicate that NAD^+^ did not bind efficiently to this receptor [43]. Here, our results indicate a regulatory role of NAD^+^ in the larval settlement and byssus thread secretion in the mussel via the SGF1.

Atrovastatin, a lipid-regulating compound, showed strong inhibitory activity against barnacle settlement [44]. It was demonstrated that atorvastatin affected the concentration of methyl farnesoate, a potential crustacean hormone, which played a key role in the lipid metabolic pathway [44,45]. Methyl farnesoate could lead to a delay or disruption in the crustacean larval development [46]. It was suggested that stimulated metabolic activities are known to deplete the energy reserves in barnacle larvae and thus retard the settlement through energy deficiency [47]. In this study, although the action mechanism of atrovastatin may be different, the results also indicate an inhibiting effect of atrovastatin in the larval settlement in the mussel, suggesting a conserved role of atrovastatin in the larval settlement in marine invertebrates, which requires investigation.

## 5. Conclusions

In this study, the in silico approach was used to screen the natural compounds’ libraries against the SGF1 responsible for the larval settlement of *M. sallei*. It was found that the targeted binding compounds could significantly affect the larval settlement, foot proteins’ gene expression, and byssus secretion of *M. sallei* adults. The laboratory bioassay results are promising, even though field trials are necessary to confirm the in silico results. Hence, this study paves the way for new experimental studies to be carried out for finding environmentally friendly antifouling agents by using SGF1-targeted compounds.

## Figures and Tables

**Figure 1 biology-13-00417-f001:**
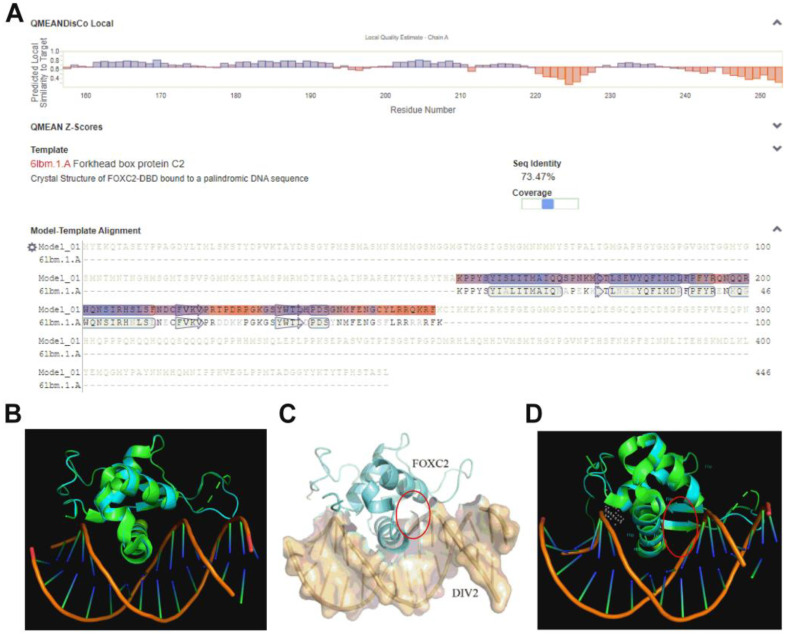
Homology model of SGF1. (**A**) SGF1 structure modeled by SWISS-MODEL. Purple: the sequence similarity is higher than 65%; orange: the sequence similarity is less than 65%. (**B**) SGF1 stacks with the template. Green: human FOXC2; blue: *M. sallei* SGF1. (**C**) Active site of the FOXC2. (**D**) Active site prediction of the SGF1. The red circle indicates the active site.

**Figure 2 biology-13-00417-f002:**
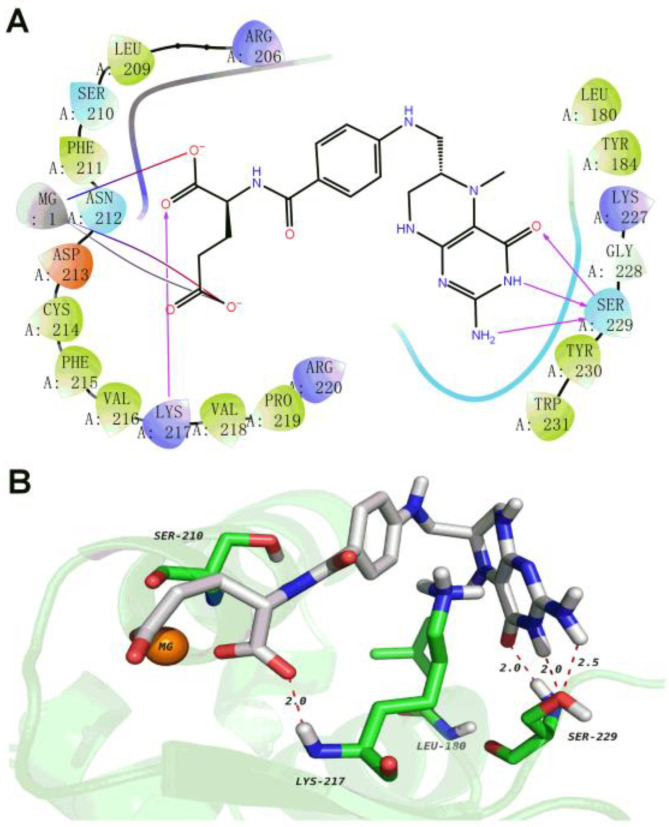
2D (**A**) and 3D (**B**) images showing binding patterns of levomefolate interacted with the SGF1. Green: carbon skeleton of SGF1; blue: nitrogen atoms; red: oxygen atoms; white: hydrogen atoms; offwhite: carbon skeleton of the compound levomefolate.

**Figure 3 biology-13-00417-f003:**
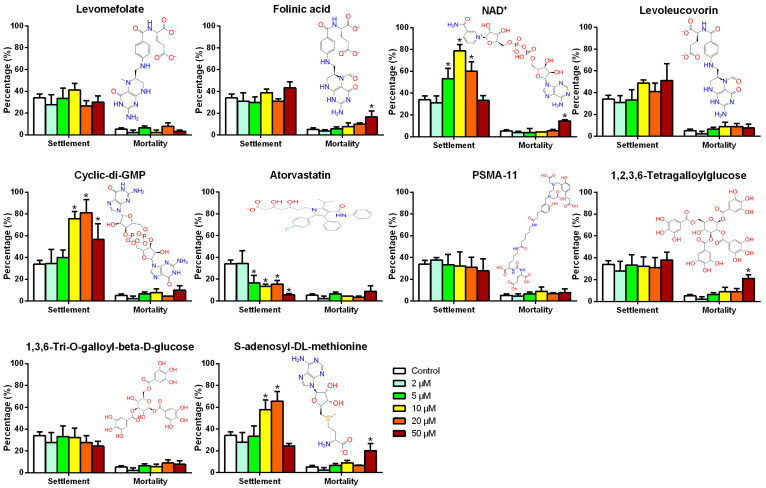
Larval settlement in response to the targeted binding compounds of SGF1. Data shown are the means of 3 replicates ± standard deviation. * denotes a significant difference between the treatments and the control (*p* < 0.05, Dunnett’s test). Control: filtered seawater with 0.5% (*v*/*v*) DMSO.

**Figure 4 biology-13-00417-f004:**
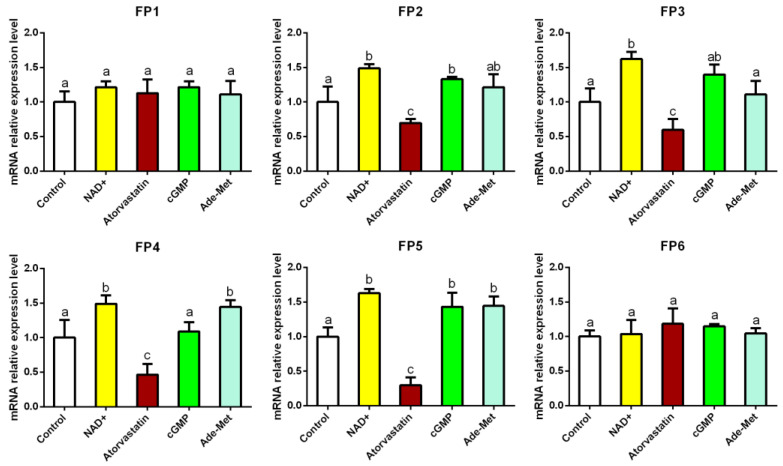
Gene expression changes regarding foot proteins in response to the targeted binding compounds of SGF1. Data shown are the means of three replicates ± standard deviation. Different letters above the bars denote significant differences among treatments (*p* < 0.05, Tukey’s test). Ade-Met: S-adenosyl-DL-methionine; cGMP: Cyclic-di-GMP. Control: filtered seawater with 0.5% (*v*/*v*) DMSO.

**Figure 5 biology-13-00417-f005:**
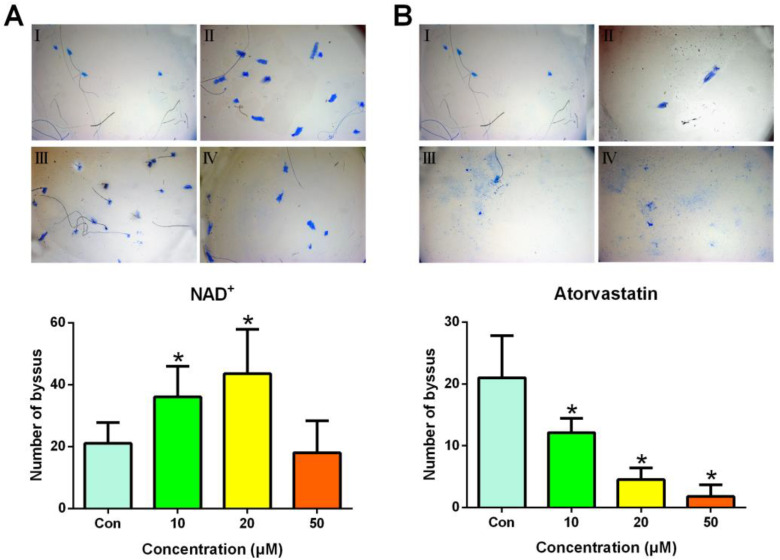
Effects of NAD^+^ (**A**) and atorvastatin (**B**) on byssal thread production of *M. sallei*. I, II, III, and IV indicate byssus secretion in response to compound at the concentrations of 0, 10, 20, and 50 μM, respectively. The blue specks indicate the adhesive plaque stained by Coomasie blue. Data shown are the means of 10 replicates ± standard deviation. * denotes a significant difference between the treatments and the control (*p* < 0.05, Dunnett’s test). Con: control, filtered seawater with 0.5% (*v*/*v*) DMSO.

**Table 1 biology-13-00417-t001:** Information for top-ranked 10 targeted binding compounds of SGF1.

Item	Catalog ID	Compound Name	Docking Score	Mw
1	HY-17383	Levomefolate (calcium)	−11.232	457.44
2	HY-B0080	Folinic acid (calcium salt pentahydrate)	−10.728	471.42
3	HY-B0445	NAD^+^	−10.011	663.43
4	HY-13667	Levoleucovorin (Calcium)	−9.938	471.42
5	HY-107780B	Cyclic-di-GMP (diammonium)	−9.671	688.40
6	HY-17379	Atorvastatin (hemicalcium salt)	−9.629	557.63
7	HY-125399	PSMA-11	−9.475	946.99
8	HY-N6006	1,3,6-Tri-O-galloyl-beta-D-glucose	−9.304	636.47
9	HY-111832	1,2,3,6-Tetragalloylglucose	−9.217	788.57
10	HY-126126	S-Adenosyl-DL-methionine	−9.110	398.44

## Data Availability

Data pertaining to this work are available from the corresponding authors upon reasonable request.

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
