# Peer review of "Silk Gland Factor 1 Plays a Pivotal Role in Larval Settlement of the Fouling Mussel Mytilopsis sallei"

_biology, 2024, doi:10.3390/biology13060417_

Round 1

Reviewer 1 Report

Comments and Suggestions for Authors

This study reported that the SGF1 plays a pivotal role in larval settlement of the fouling mussel Mytilopsis sallei. To confirm the role of SGF1, multiple targeted binding compounds of SGF1 were obtained using the high-throughput virtual screening. Furthermore, qRT-PCR was performed to reveal the expression of byssal protein genes at targeted binding compounds (NAD+ and atorvastatin). This work is interesting and provide useful information for the antifouling research. However, there are some issues that need to be addressed before the manuscript can be published.

There are many factors affecting the larva settlement. How did authors identify SGF1 from the settlement related pathways for investigation? If would be better to describe other settlement related pathways in the introduction section.

For the settlement and mortality experiment, why authors chose the concentration of 10, 20 and 50 μM?

What’s the difference of byssal proteins and foot proteins? If they are same, it is better to consistently expressed.

Why authors used DMSO in the experiment?

What’s the practical implication of this study, how to apply the findings in control the fouling mussel?

Do you think the findings of the study is specific for only the mussel M. sallei? Is it possible to apply it in other bivalve species?

The authors also need to clarify how the normalizers were chosen for the qPCR analysis…typically at least 3 normalizers in an experiment. The authors here only report beta actin which is known to not be stable in many systems. Generally the authors need to provide a much more detailed description on how the qPCR experiments were set up and controlled.

Comments on the Quality of English Language

Although there is need for clarification and English editing to improve the document, these data are clearly presented globally.

Full research article in most journal include the results and discussion section separately. In this manuscript, the results and discussion were combined together. Did authors prefer to short communication?

Author Response

We greatly appreciate the efforts of the reviewers and the editor. We have completed the modifications as requested. Below, we provide details on how we have addressed each comment (shown in blue). In the revised manuscript, the changes we have made according to the following comments are shown with red text. The line numbers included in this response letter refer to the location of the changes in the revised manuscript.

Comments and Suggestions for Authors

This study reported that the SGF1 plays a pivotal role in larval settlement of the fouling mussel Mytilopsis sallei. To confirm the role of SGF1, multiple targeted binding compounds of SGF1 were obtained using the high-throughput virtual screening. Furthermore, qRT-PCR was performed to reveal the expression of byssal protein genes at targeted binding compounds (NAD+ and atorvastatin). This work is interesting and provide useful information for the antifouling research. However, there are some issues that need to be addressed before the manuscript can be published.

  1. There are many factors affecting the larval settlement. How did authors identify SGF1 from the settlement related pathways for investigation? If would be better to describe other settlement related pathways in the introduction section.

Response: Recently, we identified that the mechanosensitive transient receptor potential melastatin-subfamily member 7 (TRPM7) channel, highly expressed in the larval foot of the mussel M. sallei, was involved in substrate exploration for settlement [1]. Further, with electrophysiological detection, transcriptomic analysis, genes, and protein expression analyses and the yeast two-hybrid technique, we identified that the TRPM7-mediated Ca2+ signal was involved in triggering the larval settlement of M. sallei through the calmodulin-dependent protein kinase kinase β/AMP-activated protein kinase/silk gland factor 1 (CaMKKβ-AMPK-SGF1) pathway. As suggested, we have explained how we identified SGF1 from the settlement related pathways for investigation (lines 82-91). In addition, we also describe other settlement related pathways in the introduction section (lines 74-79).

References:

[1] He, J.; Wang, P.; Wang, Z.; Feng, D.; Zhang, D. TRPM7-mediated Ca2+ regulates mussel settlement through the CaMKKβ-AMPK-SGF1 pathway. Int J Mol Sci 2023, 24, 5399.

  1. For the settlement and mortality experiment, why authors chose the concentration of 10, 20 and 50 μM?

Response: We chose the concentrations of 2, 5, 10, 20, and 50 μM for the larval settlement experiments. It showed that there were no significant difference between the treatments and the control at the concentrations of 2, and 5 μM, except for NAD+ and atorvastatin at 5 μM. To simplify the results, we did not provide these results before. After our consideration, we believe it is necessary to provide these data. We have modified it in the revision.

  1. What’s the difference of byssal proteins and foot proteins? If they are same, it is better to consistently expressed.

Response: Sorry for the confusing. In this study, byssal proteins are the same with foot proteins. As suggested, we have replaced byssal proteins with foot proteins in the revised manuscript.

  1. Why authors used DMSO in the experiment?

Response: Some compounds are less water-soluble, such as atorvastatin. To improve the water solubility of the compounds, DMSO was used. The previous study [1] and the pilot study showed there was no significant effect of 0.5% (v/v)DMSO on larval settlement and survival in M. sallei. Thus, filtered seawater with 0.5% DMSO was used.

Reference:

[1] He, J.; Wu, Z.; Chen, L.; Dai, Q.; Hao, H.; Su, P.; Ke, C.; Feng, D. Adenosine triggers larval settlement and metamorphosis in the mussel Mytilopsis sallei through the ADK-AMPK-FoxO pathway. ACS Chem Biol 2021, 16, 1390-1400.

  1. What’s the practical implication of this study, how to apply the findings in control the fouling mussel?

Response: In recent years, using natural products as environmentally friendly antifouling agents to combat foulers is a promising approach. However, the search for effective antifouling agents is hampered by the lack of well-defined conserved molecular targets responsible for regulating larval settlement in fouling organisms.In this study, we confirmed the targeted binding compounds of SGF1 could affect mussel settlement. It is promising to search for effective antifouling agents against SGF1 using the high-throughput virtual screening. In the future, more targeted binding compounds of SGF1, which could significantly inhibit larval settlement would be obtained. Field trials that incorporated these compounds as the antifouling agents into paints would be carried out. Hence, this study paves way for new experimental studies to be carried out for finding environmentally friendly antifouling agents by using SGF1-targeted compounds. 

  1. Do you think the findings of the study is specific for only the mussel sallei? Is it possible to apply it in other bivalve species?

Response: We think the findings of the study is not specific for only the mussel M. sallei. Recently, we also tested the effects of larval settlement and byssus threads Secreting in response to the targeted binding compounds of SGF1 in other bivalves, such as the mussel Perna viridis, and the scallop Chlamys nobilis, which shows similar results with M. sallei. Therefore, we think the findings are suitable for some byssus-secreting bivalves.

  1. The authors also need to clarify how the normalizers were chosen for the qPCR analysis…typically at least 3 normalizers in an experiment. The authors here only report beta actin which is known to not be stable in many systems. Generally the authors need to provide a much more detailed description on how the qPCR experiments were set up and controlled.

Response: Thanks for the suggestion. In our previous studies, we have proved beta actin was stable in qPCR analysis for larval development in M. sallei [1, 2]. As suggested, we have modified the qPCR analysis in the revised manuscript.

Reference:

  • He, J.; Wu, Z.; Chen, L.; Dai, Q.; Hao, H.; Su, P.; Ke, C.; Feng, D. Adenosine triggers larval settlement and metamorphosis in the mussel Mytilopsis salleithrough the ADK-AMPK-FoxO pathway. ACS Chem Biol 2021, 16, 1390-1400.

[2] He, J.; Wang, P.; Wang, Z.; Feng, D.; Zhang, D. TRPM7-mediated Ca2+ regulates mussel settlement through the CaMKKβ-AMPK-SGF1 pathway. Int J Mol Sci 2023, 24, 5399.

Comments on the Quality of English Language

Although there is need for clarification and English editing to improve the document, these data are clearly presented globally.

Response: Thanks, we have made moderate editing to improve the document.

Full research article in most journal include the results and discussion section separately. In this manuscript, the results and discussion were combined together. Did authors prefer to short communication?

Response: Thanks. As suggested, we have modified the results and discussion section separately.

Reviewer 2 Report

Comments and Suggestions for Authors

In this manuscript, the authors explored the response to the targeted binding compounds of SGF1, involved in triggering the larval settlement in the mussel Mytilopsis sallei. The manuscript is well-written, however, there are some questions need to be addressed.

(1) How did the authors determine larval settlement in Fig.3?

(2) Why did the authors use only NAD+ to observe the expression changes of foot proteins in Fig.4? Did other compounds with inducing effects on larval settlement not change the expression of FPs? FPs form the final adhesive plaque on the substrates in mussels adhesion. How do FPs correlate with the settlement process?

(3) In Fig.5, mussels with the size of 8-12 mm were used. Why did the authors not use pediveliger larvae in Figs.3 & 4? It was not clear to me what the photos in Fig.5 mean. What are those blue things? How long are byssus threads produced in 24 h of incubation?

Author Response

We greatly appreciate the efforts of the reviewers and the editor. We have completed the modifications as requested. Below, we provide details on how we have addressed each comment (shown in blue). In the revised manuscript, the changes we have made according to the following comments are shown with red text. The line numbers included in this response letter refer to the location of the changes in the revised manuscript.

Comments and Suggestions for Authors

In this manuscript, the authors explored the response to the targeted binding compounds of SGF1, involved in triggering the larval settlement in the mussel Mytilopsis sallei. The manuscript is well-written, however, there are some questions need to be addressed.

  • How did the authors determine larval settlement in Fig.3?

Response: In our previous study, we have described how to determine larval settlement (He et al. 2016). According to the descriptions in the relevant literature (García-Lavandeira et al. 2005; Grant et al. 2013), larval attachment was confirmed by crawling with an extended foot or attaching by byssus while the velum is absorbed (Figure 1b,c); larval metamorphosis was confirmed by loss of the velum, the appearance of mature gill filaments and the appearance of a shell morphologically similar to that of adult mussels (Figure 1d,e); larvae that showed no signs of movement of the velum, foot, or gut while the tissue inside the shell became ulcerated were considered dead (Figure 1f). In this study, the attachment and metamorphosis, both processes are referred, as settlement. The determination of larval settlement has been described in the revision (lines 144-148).

Figure 1. M. sallei pediveligers that were swimming (a, competent to settle and metamorphose), settled (b, vertical view; c, lateral view), metamorphosed (metamorphosed into plantigrade: d, vertical view; e, lateral view) and dead (f). Abbreviations: b, byssus; f, foot; v, velum. Arrows indicate gill filaments. Scale bar: 100 μm.

References:

He, J.; Qi, J.F.; Feng, D.Q.; Ke, C.H. Embryonic and larval development of the invasive biofouler Mytilopsis sallei (Récluz, 1849)(Bivalvia: Dreissenidae). J Mollus Stud 2016, 82, 23–30.

García-Lavandeira, M.; Silva, A.; Abad, M.; Pazos, A.J.; Sánchez, J.L.; Pérez-Parallé, M.L. Effects of GABA and epinephrine on the settlement and metamorphosis of the larvae of four species of bivalve molluscs. J Exp Mar Biol Ecol 2005,316, 149–56.

Grant, M.N.; Meritt, D.W.; Kimmel, D.G. Chemical induction of settlement behavior in larvae of the eastern oyster Crassostrea virginica (Gmelin). Aquaculture 2013 402–403, 84–91.

  • Why did the authors use only NAD+ to observe the expression changes of foot proteins in Fig.4? Did other compounds with inducing effects on larval settlement not change the expression of FPs? FPs form the final adhesive plaque on the substrates in mussels adhesion. How do FPs correlate with the settlement process?

Response: We also analysed the expression changes of foot proteins in response to other compounds with inducing effects. The other compounds (cyclic-di-GMP and S-adenosyl-DL-methionine) with inducing effects on larval settlement also changed the expression of FPs (Figure 2). Among the three inducing compounds, NAD+ showed the biggest effect on expression of FPs, which could change expression of FP2, FP3, FP4, and FP5. Therefore, we used only NAD+ to observe the expression changes of foot proteins in Fig.4. After our consideration, we believe it is necessary to provide all these data. We have modified it in the revision.

At the early stage, larval settlement is confirmed by crawling with an extended foot; At the late stage, larval settlement is confirmed by attaching on the substrates by byssus, which is formed by PFs. Thus, the completion of larval settlement is mediated by the expression of PFs.

Figure 2. Genes expression changes of foot proteins in response to the targeted binding compounds of SGF1.

(3) In Fig.5, mussels with the size of 8-12 mm were used. Why did the authors not use pediveliger larvae in Figs.3 & 4? It was not clear to me what the photos in Fig.5 mean. What are those blue things? How long are byssus threads produced in 24 h of incubation?

Response: In Fig.5, byssus threads secreting in response to the targeted binding compounds of SGF1 were determined. We did not use the pediveliger larvae, because the shell length of pediveliger larvae was about 230 μm (He et al. 2016). It is difficult to observe the numbers of byssus threads produced by each pediveliger larva in response to NAD+ and atorvastatin. Thus, mussels with the size of 8-12 mm were used. In Fig.5, the blue spots indicate the adhesive plaque of the byssus threads, which could be stained by Coomassie blue. The number of the blue spots means the numbers of byssus threads produced by each mussel. Sorry for the confusing. We have improved the description of the methods in the the revised manuscript. The length of the byssus threads was about 1-2 cm in 24 h of incubation.

He, J.; Qi, J.F.; Feng, D.Q.; Ke, C.H. Embryonic and larval development of the invasive biofouler Mytilopsis sallei (Récluz, 1849)(Bivalvia: Dreissenidae). J Mollus Stud 2016, 82, 23-30.

Round 2

Reviewer 1 Report

Comments and Suggestions for Authors

I think the revised manuscript can be accepted for publication.

Comments on the Quality of English Language

Ok.

Reviewer 2 Report

Comments and Suggestions for Authors

The manuscript has been revised well. I think this will be acceptable.